

# The impact of feature selection on one and two-class classification performance for plant microRNAs

Waleed Khalifa[1,2], Malik Yousef[1,2], Müşerref Duygu Saçar Demirci[3] and Jens Allmer[3,4]

[1] Computer Science, The College of Sakhnin, Sakhnin, Israel
[2] The Institute of Applied Research- The Galilee Society, Shefa Amr, Israel
[3] Molecular Biology and Genetics, Izmir Institute of Technology, Urla, Izmir, Turkey
[4] IZTEKGEB, Bionia Incorporated, Urla, Izmir, Turkey

Corresponding author
Malik Yousef,
malik.yousef@gmail.com

## ABSTRACT

MicroRNAs (miRNAs) are short nucleotide sequences that form a typical hairpin structure which is recognized by a complex enzyme machinery. It ultimately leads to the incorporation of 18–24 nt long mature miRNAs into RISC where they act as recognition keys to aid in regulation of target mRNAs. It is involved to determine miRNAs experimentally and, therefore, machine learning is used to complement such endeavors. The success of machine learning mostly depends on proper input data and appropriate features for parameterization of the data. Although, in general, two-class classification (TCC) is used in the field; because negative examples are hard to come by, one-class classification (OCC) has been tried for pre-miRNA detection. Since both positive and negative examples are currently somewhat limited, feature selection can prove to be vital for furthering the field of pre-miRNA detection. In this study, we compare the performance of OCC and TCC using eight feature selection methods and seven different plant species providing positive pre-miRNA examples. Feature selection was very successful for OCC where the best feature selection method achieved an average accuracy of 95.6%, thereby being ∼29% better than the worst method which achieved 66.9% accuracy. While the performance is comparable to TCC, which performs up to 3% better than OCC, TCC is much less affected by feature selection and its largest performance gap is ∼13% which only occurs for two of the feature selection methodologies. We conclude that feature selection is crucially important for OCC and that it can perform o*n par* with TCC given the proper set of features.

## INTRODUCTION

Gene regulation is of prime importance in all living organisms and there are multiple levels at which gene expression can be modulated. MicroRNAs (miRNAs) play a role in post-transcriptional gene regulation (*Erson-Bensan*, *2014*) and, among other functions, fine-tune the amount of translated protein product (*Saçar & Allmer*, *2013b*). Mature miRNAs are short nucleotide sequences discovered about two decades ago

(*Lee, Feinbaum & Ambros*, *1993*). From databases which host miRNAs like miRBase (*Kozomara & Griffiths-Jones*, *2011*) it can be gleaned that miRNAs exist in a wide range of organisms ranging from viruses (*Grey*, *2015*) to plants (*Yousef, Allmer & Khalifa*, *2016*). It has also been proposed that the plant miRNA system may have evolved independently (*Chapman & Carrington*, *2007*) and some organisms like yeasts also display differences to the canonical pathway (*Ender & Meister*, *2010*). Any regulatory element itself may be miss-regulated and miRNAs are no exception, and therefore, have been implicated in, for example, human diseases (*Alural et al.*, *2014*; *Alural et al.*, *2015*) and in plant stress response (*Zhang et al.*, *2010*). While miRNAs may lead to inter-kingdom communication in special cases (*Bağcı & Allmer*, *2012*), it is not likely that there is extensive communication among eukaryotes (*Bağcı & Allmer*, *2016*). Experimentally detected and/or validated miRNAs are available in databases such as miRBase (*Griffiths-Jones et al.*, *2008*) and miRTarBase (*Hsu et al.*, *2011*). MicroRNAs' effect can only be established when it is co-expressed with its targets (*Saçar & Allmer*, *2013b*), which complicates experimental analysis since only a fraction of the genome is expressed at a given time, in a tissue, or in response to stress conditions; and testing all conditions experimentally is elusive. Additionally, such an analysis needs to be performed on transcript and protein level, concurrently, over multiple time points to establish a causative relationship. Therefore, it seems impossible to experimentally detect all possible miRNAs of any higher eukaryotic organism (*Yousef et al.*, *2008*; *Ding, Zhou & Guan*, *2010*; *Wu et al.*, *2011*; *Ritchie, Gao & Rasko*, *2012*). Moreover, it has become clear that even among the experimentally validated miRNAs in miRBase and mirTarBase, there may be dubious examples (*Saçar, Hamzeiy & Allmer 2013*). Therefore, carefully designed computational experiments are required to complement experimental approaches for miRNA detection.

Many computational approaches to miRNA detection have been proposed and most of them derive numerical features (*Sacar & Allmer*, *2013*) to describe a pre-miRNA and then use machine learning to establish a model for miRNA identification (*Allmer & Yousef*, *2012*; *Yousef, Allmer & Khalifa*, *2015*; *Saçar & Allmer*, *2013a*; *Saçar & Allmer*, *2014*; *Allmer*, *2014*). Of these approaches, most, with few exceptions (*Yousef et al.*, *2008*; *Yousef, Allmer & Khalifa*, *2015*; *Koski et al.*, *2005*), employ two class classification; the latter has been compared previously (*Saçar & Allmer*, *2013a*; *De On Lopes, Schliep & De Lf de Carvalho*, *2014*). Classification in machine learning depends on positive examples for training the classifier in case of one-class classification (OCC) and additionally on negative data in case of two-class classification (TCC). The negative data, however, proves to be difficult to establish (if not impossible), so that all negative datasets currently in use are based on arbitrary selection of examples from parts of a genome deemed not miRNA genic or from randomly generated sequences. While both approaches are questionable, they present the only alternative to using OCC and in the absence of proper benchmark data need to be used for TCC (*Allmer*, *2012*). Since OCC only needs examples for the target class (here positives), it can obliterate the need to define artificial negative examples (*Manevitz & Yousef*, *2002*; *Manevitz & Yousef*, *2007*) and can be used to differentiate between target and unknown class. We have recently analyzed the use of OCC for miRNA detection in plants and found that it was competitive in comparison to TCC although the analysis was unduly

biased towards TCC (*Yousef, Allmer & Khalifa*, *2016*). Our previous study also showed that among the hundreds of features proposed for miRNA parameterization (*Sacar & Allmer*, *2013*) some are more discriminative than others. Since feature selection is NP-hard (*Amaldi & Kann*, *1998*), selecting the best subset from more than 1,000 features on a per dataset basis is not achievable. Feature selection has been investigated before, but mostly for TCC (*Paul, Magdon-Ismail & Drineas*, *2015*; *Guyon et al.*, *2002*; *Ahsen et al.*, *2012*), while only little has been done for OCC (*Lorena, Carvalho & Lorena*, *2015*; *Xuan et al.*, *2011a*; *Hall et al.*, *2009*). In this study, we used different feature selection approaches and compared their effectiveness for OCC and TCC classification performance.

Both machine learning approaches, OCC and TCC, benefit from feature selection. While feature selection is essential for OCC and a difference of about 30% accuracy can be observed, the maximum difference for TCC is ∼10%. Moreover, for TCC 7 out of 8 feature selection methods lead to accuracy greater than 90% whereas such high accuracy was only achieved for two methods when using OCC. For the LIG feature selection method, intended as a negative control, both classifiers display lowest performance but TCC is about 20% better than OCC. With increasing accuracy (i.e., better feature selection for OCC), the accuracy for TCC also increases; except for the SFC which is best for OCC but only third best for TCC. While the performance difference for LIG is large, it decreases with the use of better feature selection methods. TCC is only 3% better when the SFC feature selection method is considered, which provided the best performance for OCC. A difference in performance among plant species was observed for both classifiers, but for TCC it was about 5% whereas for OCC it was 15%. In conclusion, feature selection is essential for OCC, but does not affect TCC as much. We propose that due to the lack of true negative data, more focus should be put on the further development of OCC approaches to pre-miRNA detection.

## MATERIALS AND METHODS

### Data

Positive examples for pre-miRNAs from selected plant species were downloaded from miRBase (*Griffiths-Jones et al.*, *2008*) (Releases 20 and 21). *Glycine max* (gma), *Zea mays* (zma), *Sorghum bicolor* (sbi), *Physcomitrella patens* (ppt), *Arabidopsis thaliana* (ath), *Populus trichocarpa* (ptc), and *Oryza sativa* (osa) make up the positive dataset. Negative examples for miRNAs consisted of 980 pseudo pre-miRNAs from the PlantMiRNAPred dataset (*Xuan et al.*, *2011b*). For these data, all pre-miRNA features were calculated as described previously (*Sacar & Allmer*, *2013*; *Yousef, Allmer & Khalifa*, *2015*; *Saçar, Bağcı & Allmer*, *2014*). We chose plant pre-miRNAs with large amount of pre-miRNA examples and from different clades for this study. Additionally, plant miRNAs have not been investigated as extensively as metazoan miRNAs which adds to the reason to choose plant pre-miRNAs.

### One class classification

For one-class classification the DDtools (*Tax*, *2015*) implementation of an OCC was utilized. A 100-fold Monte Carlo cross validation (*Xu & Liang*, *2001*) was performed using randomly sampled 90% of the positive data for training and 10% for testing. Moreover, the pseudo

negative sequences were injected as unknown class during testing. We employed *k*-means in this study as previously described (*Yousef, Allmer & Khalifa*, *2016*) since it performed well in respect to OCC although it is a clustering algorithm. During learning, labeled examples are clustered (miRNAs and unknown) and during testing and in prediction, the label of the closest cluster is assigned to the sample.

## Two class classification

Support Vector Machines (SVMs) are used for machine learning and were first proposed by *Vapnik (1995)*. In bioinformatics and in the field of pre-miRNA detection, SVMs have been used (*Ding, Zhou & Guan*, *2010*; *Wu et al.*, *2011*; *Xuan et al.*, *2011b*; *Ng & Mishra*, *2007*). Here, the WEKA library (*Gewehr, Szugat & Zimmer*, *2007*) SVM implementation which is based on LibSVM (*Chang & Lin*, *2011*) was utilized. The radial basis function was set to a gamma value of 0.7 and the cost parameter was chosen to be 4.0 and the normalization option was set to true. Any machine learning algorithm needs initial training and we performed a 10 fold Monte Carlo cross validation (*Xu & Liang*, *2001*) during learning, by employing random sampling using 90% of the data for training and 10% for testing.

## Feature selection strategies

Feature selection has been shown to be an NP-hard problem and, therefore, other approximate feature selection strategies are being developed. In machine learning for pre-miRNAs, more than 1,000 features have been proposed which makes feature selection especially hard. To investigate the impact of feature selection on model performance for OCC and TCC, four negative and four positive feature selection methods were designed. Previously, we found that a set of 50–100 features may be sufficient for successful pre-miRNA detection (*Sacar & Allmer*, *2013*). Using more than 50 features increases the likelihood that the feature set contains some features which may conceal differences among feature selection methods. Therefore, a feature set size of 50 was selected for model training in this study.

We have previously performed feature selection for OCC (*Yousef et al.*, *2016*) using similar feature selection methods as we propose here, but it is important to compare the impact between OCC and TCC.

Eight feature selection methods were devised and four of them were expected to lead to low performance while the remaining methods were thought to perform well. The former were selecting features with low information gain (LIG), random feature selection (RFS), selecting random feature from feature clusters (RFC), and selecting features from clusters (SFC). The latter were selecting features with high information gain (HIG), selecting the highest information gain from feature clusters (HIC), zero-norm feature selection (ZNF), and Pearson correlation-based feature selection (PCF).

All feature selection methods except for the last two were performed using KNIME (*Berthold et al.*, *2009*) and the selected features are available in Table S1; information on how to calculate them are provided in File S1. The workflows for our feature selection methods, developed in KNIME, are available for download from our website: http://bioinformatics.iyte.edu.tr/supplements/featsel.

In order to calculate LIG and HIG, for each dataset, the information gain (IG) among features was established (using KNIME's InformationGainCalculator node) and the 50 features with lowest IG (LIG) or highest IG (HIG) were selected. For RFS, 50 random features were selected using the Row Sampling node in KNIME. To establish RFC, features were clustered using WEKA $k$-Means implementation in KNIME ($k = 100$). From each cluster a random feature was selected and from the 100 random features the final set of 50 features was selected randomly (KNIME's Row Sampling approach). For SFC, clustering was performed as for RFC. Clusters were ordered by number of cluster members (largest to smallest) and the 50 features were chosen from the top. To derive HIC, the same clustering approach as for RFC and SFC was taken, but the features in each cluster were ranked according to IG and the best one was selected. The selected 100 features were again ranked using IG and the best 50 were selected. ZNF is defined to be the non-zero values for all feature vectors of positive examples. Among the non-zero ones, the 50 features with highest sum of values were selected. PCF was established according to *Lorena, Carvalho & Lorena (2015)* and after Pearson clustering the features with lowest correlation score were retained. Feature selection was performed on a per species basis which led to the selection of different features (Table S1). Combined feature selection uses the occurrence of selected features among the seven selected species and five mixed datasets in respect to the top 100 features. The Features were ranked according to their frequency and top 50 were selected (Table S1).

## RESULTS AND DISCUSSION

Eight feature selection methods were designed and they were applied to seven plant datasets. For each dataset OCC (100) and TCC (10) models were established using Monte Carlo cross validation (MCCV). Feature selection was performed on a per plant dataset basis. The 50 features selected varied to some extend and, therefore, we defined another feature set (indicated by 'comb') which was created by selecting the features ordered by decreasing incidence based on the individual selections. The selected features are provided in Table S1 by their acronyms which are explained in more detailed in our previous studies (*Sacar & Allmer*, 2013; *Saçar & Allmer*, 2013a).

We applied the eight feature selection methods to the seven plant species' datasets individually and recorded the model performance. Figure 1 shows the average model performance (OCC: 100, TCC: 10 fold cross validation) for the best feature selection method we found (SFC) and the worst one (LIG).

Sensitivity was the performance measure most affected for both machine learning approaches (Fig. 1). For TCC the average accuracy among plant species dropped about 10% between SFC and LIG while it dropped about 30% for OCC. The results for the remaining six feature selection methods are presented in Table S2.

The impact on using combined feature selection for SFC and LIG is quite similar to individual feature selection (Fig. 2). The combined features were not calculated for PCF and ZNF since combination of features was not supported by our workflow in this case. Overall accuracy is slightly reduced for the combined feature selection by on average 1% (OCC) and 2% (TCC) when compared to individual feature selection.
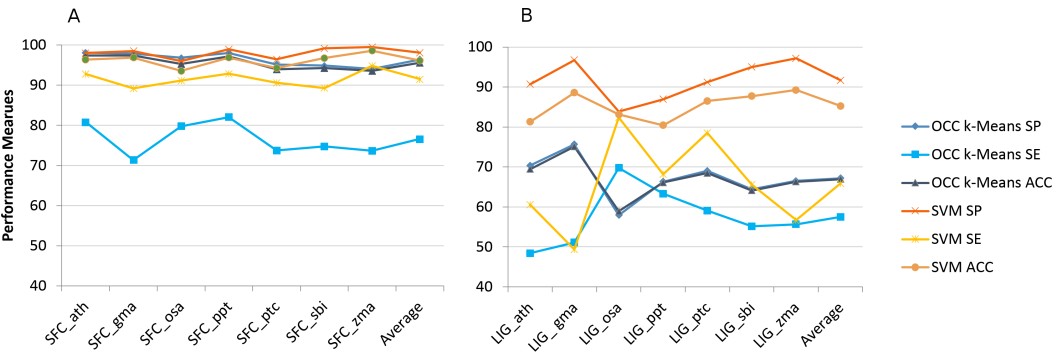

**Figure 1  Best (SFC) versus worst (LIG) feature selection method on per species feature selection.** Average model performance for SFC feature selection method for selected plant species (A) and LIG feature selection model (B). OCC performance is in blue tone and SVM is toned orange. SP, specificity; SE, sensitivity; and ACC, accuracy. Lines between points do not convey meaning, but were used to simplify visual tracking. Table S2 contains further information for all feature selection methods as well as standard deviations.

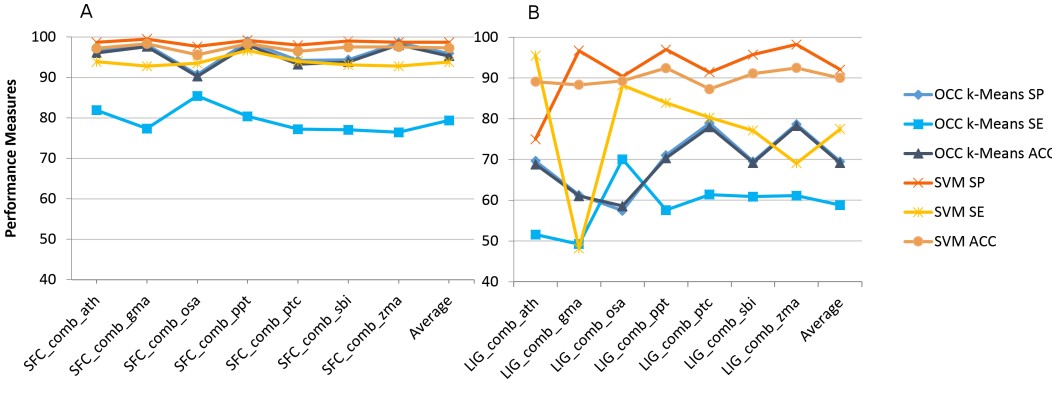

**Figure 2  Best (SFC) versus worst (LIG) feature selection method on consensus feature selection.** Average model performance for SFC feature selection method using combined feature set for selected plant species (A) and LIG feature selection model using combined feature set (B). OCC performance is in blue tone and SVM is toned orange. SP, specificity; SE, sensitivity; and ACC, accuracy. Lines between points do not convey meaning, but were used to simplify visual tracking. Table S2 contains further information for all feature selection methods as well as standard deviations.

The performance analysis of the remaining feature selection methods are presented in Table S2. In order to compare the performance of all feature selection methods for the two machine learning approaches, the average model accuracy was plotted (Fig. 3). It is striking that for most (six out of eight) TCC performance results the accuracy is above 95% for all plant species.

For OCC, the performance is best for SFC where, on average, for plant species it achieves more than 95% accuracy (Fig. 3). All other feature selection methods do not lead to high performing models with HIC being the second best, followed by RFS and PCF. For most

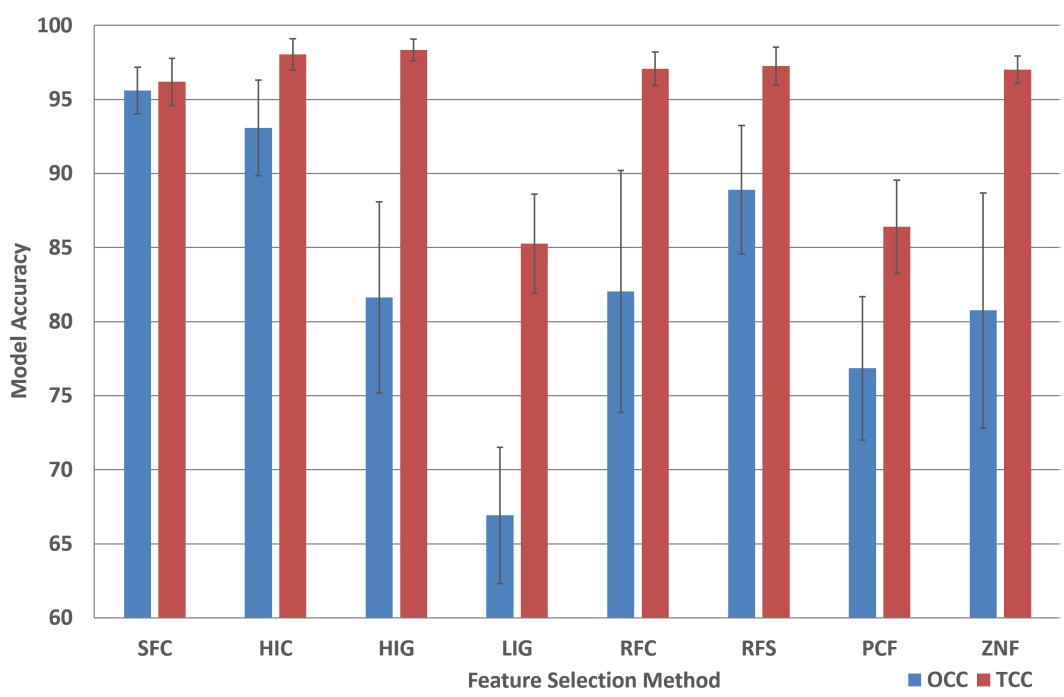

**Figure 3** Model accuracy comparison between OCC and TCC in respect to feature selection method. The average accuracy of OCC and TCC models created for selected plant species using eight different feature selection methods and their standard deviation. Data and figure are available for closer analysis in Table S2.

feature selection methodologies the accuracy among plant species is quite similar for TCC, but for OCC the differences are much larger.

In order to compare the variance in performance between OCC and TCC, the difference between TCC and OCC accuracy was calculated ($TCC_{ACC}$–$OCC_{ACC}$) and is presented in Fig. 4. Positive values signify better performance of TCC.

The most accurate OCC model is on the left (SFC) and it is seen that TCC is outperformed by OCC on several plant species (ath, gma, osa, and ppt). Figure 4 shows that OCC is more affected by feature selection than TCC and, therefore, with increasing effectiveness of the feature selection methodology, the difference between classifiers diminishes. For improper feature selection it can reach up to about 30%, whereas it drops to almost similar performance for the best feature selection method in this study (SFC, $\sim 0.6\%$ on average).

## CONCLUSIONS

Many general purpose feature selection methods have been described or used in bioinformatics (*Saeys, Inza & Larrañaga*, *2007*). For OCC feature selection nothing has been done in the area of pre-miRNA detection while one study investigated feature selection based on OCC for mature miRNA prediction (*Xuan et al.*, *2011a*). When considering two class classification of pre-miRNAs SVM recursive feature elimination (RFE) has been used (*Shu et al.*, *2015*). *Meng et al.*, *(2014)* also used RFE, but modified it and compared to principal component analysis (PCA), correlation-based feature subset selection (CFS),

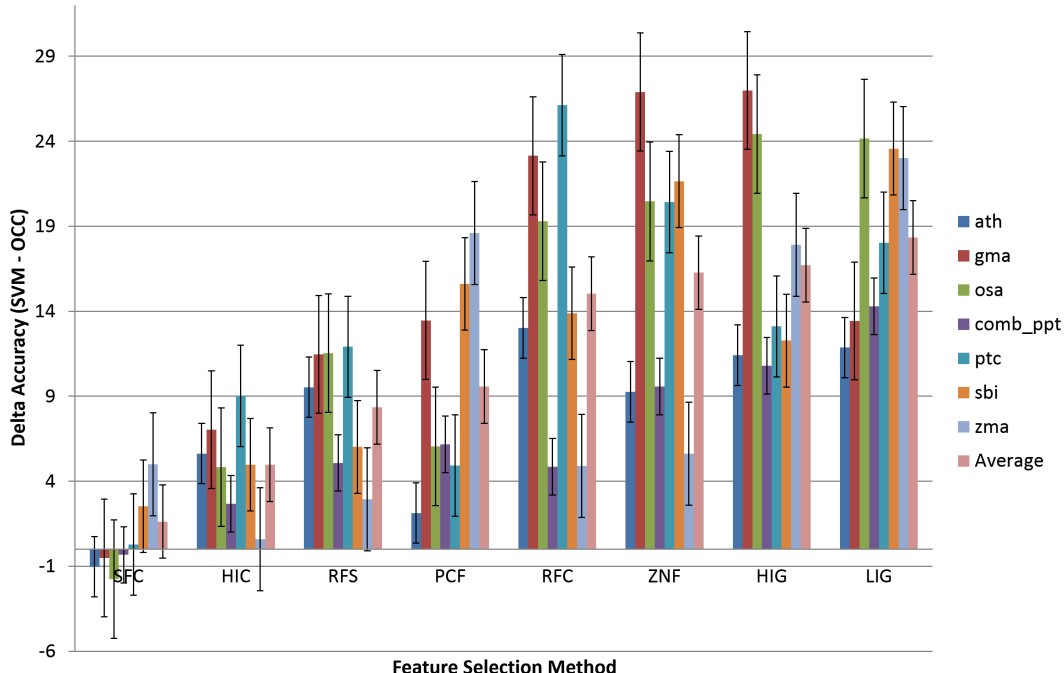

**Figure 4** **Comparison of the effect of feature selection on two-class versus one-class classification.**
Eight feature selection methodologies applied to OCC and TCC. The difference in accuracy between TCC
and OCC is presented. The groups are sorted by increasing average difference. Results are presented on a
per species basis. The 'Average' averages the OCC or TCC performance among species. Data and figure are
available for closer analysis in Table S2.

and not using any filtering. They report the best accuracy for SVM using their back
SVM-RFE FS with 97.2% closely followed by PCA using SVM with 97.0% accuracy. One
approach used genetic algorithm in combination with information gain and also taking
into account feature redundancy for FS and achieved almost 99.5% accuracy, alas on a
limited dataset (*Xuan et al.*, *2011c*). These competing methods using different strategies
for FS in pre-miRNA detection do not refer to OCC. However, they clearly show that
feature selection has a large impact on model performance. The previous methodologies
used correlation among features or feature redundancy for FS but did not put a clear
focus on the correlation issue. We, therefore, devised eight feature selection methodologies
with a focus on feature correlation and applied them to several plant miRNA datasets.
Feature selection was performed on a per plant species basis, but we also investigated the
combined feature set using the features shared among species; both of which were not
done in previous approaches. Our SFC feature selection methodology was particularly
successful and there was no great difference for feature selection on a per plant basis or
when combined (Figs. 1A and 2A). As expected, the LIG methodology did not perform well
at all and was intended as a negative control. However, the SVM learner was not nearly as
much affected as the OCC one (Figs. 1B and 2B) although sensitivity was strongly affected
for both learners.

Of the eight feature selection methods tested in this study, only 3 show good performance
for OCC (SFC, HIC, and RFS; Fig. 3) while only two did not seem applicable for SVM (LIG

and PCF; Fig. 3). For most feature selection methods average SVM performance is above 95% while OCC performance is generally below 90% (Fig. 3).

It is instructive to analyze the performance difference between SVM and OCC. Figure 4 shows the performance difference and it can be seen that for most feature selection methods SVM performs better than OCC (positive values; Fig. 3). However, the SFC feature selection method which is among the best for SVM clearly performs best for OCC and the latter can surpass the SVM performance for several of the selected plant species.

From this study it can be concluded that the more successful the feature selection the less difference between OCC and TCC model performance and the better the overall model performance. Thus we conclude, that in the absence of missing negative data OCC should be used and, therefore, additional feature selection strategies should be tried to improve its performance.

### Funding
The work was supported by the Scientific and Technological Research Council of Turkey (grant number 113E326) to JA. The funders had no role in study design, data collection and analysis, decision to publish, or preparation of the manuscript.

### Grant Disclosures
The following grant information was disclosed by the authors:
Scientific and Technological Research Council: 113E326.

### Competing Interests
The authors declare there are no competing interests.

### Author Contributions
- Waleed Khalifa conceived and designed the experiments, performed the experiments, prepared figures and/or tables, reviewed drafts of the paper.
- Malik Yousef and Jens Allmer conceived and designed the experiments, performed the experiments, analyzed the data, contributed reagents/materials/analysis tools, wrote the paper, prepared figures and/or tables, reviewed drafts of the paper.
- Müşerref Duygu Saçar Demirci performed the experiments, prepared figures and/or tables, reviewed drafts of the paper.

### Data Availability
All data has been supplied as Supplemental Information.

### Supplemental Information
Supplemental information for this article can be found online at http://dx.doi.org/10.7717/peerj.2135#supplemental-information.

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
