# Peer review of "The impact of feature selection on one and two-class classification performance for plant microRNAs"

_PeerJ, doi:10.7717/peerj.2135_

## Round 0.1 · original submission · Major Revisions

As the reviewers indicate, more details are needed to for the methods and the parameters chosen.

Reviewer 1 ·

Basic reporting

"No Comments".

Experimental design

- In Section Feature Selection Strategies, authors describe: "Previously, we found that 50 features may be sufficient for successful pre-miRNA detection [21] and, therefore, selected 50 features for model training in this study. We have previously described these feature selection methodologies (Yousef, Saçar, Demirci, Khalifa, and Allmer; submitted), but would like to give a brief summary here."

This part is not very clear, probably because few information is given. I would reference Supplementary Table 1 at this point, because by looking at this table reader can realize that “50 features” relates to a good “feature set size” rather than a good “feature set” to perform pre-miRNA prediction.


- In the legends of Figure 1 and 2, authors report results for “OCC k-means”, but the use of k-means wasn’t described at any point of the text. This was confusing to me, since clustering was only mentioned when RFC, SFC and HIC feature selection method were briefly described. The relation of k-means with OCC should be made clearer in the text.


- Is the “average” column in figures 3 and 4 the average performance for all the species, or all the species plus the combined features? The meaning of this column should be better explained in the text or figure captions.

- In Figure 3, I suggest including a “shaded” background either for OCC or TCC, in order to make it clearer for interpretation, so that reader can more easily identify the OCC or the TCC columns in the figure.

Validity of the findings

- I suggest performing a statistical test to check if difference in performance is meaningful, both for OCC vs TCC comparison, and among different feature selection methods.

- Moreover, standard deviations across the 100-fold or 10-fold cross validation should be reported along with the means (at least in the supplementary tables).

Additional comments

This paper is well-written, clearly organized and provides enough background for the reader to understand the problem under consideration. Also, it tackles a very important issue related to classification of pre-miRNAs, specifically, the lack of negative examples and the need to investigate efficient strategies to deal with this limitation. In this sense, comparison studies like the one held in this manuscript are important to elucidate efficient methodologies and computational approaches to assist in the advance of this research problem. My specific comments for this paper aim to help make it even more comprehensible for future readers.

Reviewer 2 ·

Basic reporting

No Comments

Experimental design

No Comments

Validity of the findings

No Comments

Additional comments

The main goal of this work is to demonstrate the impact of using different feature selection methods on the classification performance of microRNAS. The authors evaluate the use of eight feature selection methods but they do not explain details from these methods. In line 102 is highlighted that the feature selection methodologies are under review by another journal: "We have previously described these feature selection methodologies (Yousef, Saçar 103 Demirci, Khalifa, and Allmer; submitted)". In my opinion, it is not possible to correctly evaluate the accuracy of the results without knowing details of the implemented methods. The authors should explain the proposed methods and also explain how the proposed techniques advances in this field of research and/or contributes in something new to the literature. An appropriate section of 'related work' in this context should present a focused discussion of relevant works in this field. How is this paper contributing to the state-of-the-art? What are the advantages (or disadvantages) of using these feature selection methods, in comparison to other approaches? What are the specific components of each feature selection method responsible for these successful results? The content and technical quality of the paper require significant improvements. Whereas that feature selection is an NP-hard problem, I would like to see any evaluation of the execution time/computational effort required by the proposed methods.

Reviewer 3 ·

Basic reporting

The manuscript presents a comparison among feature selection methods for plant microRNAs classification task. The microRNAs identification is an important issue in bioinformatics research field. However, the manuscript must be improved in order to describe how the work fits into the broader field of knowledge.
It is mentioned in the manuscript various methods and its parameters, however it is not justified these choices. There is mention by the authors: "We have previously described these feature selection methodologies (Yousef, Saçar Demirci, Khalifa, and Allmer; submitted), but would like to give a brief summary here.". However, the description of adopted methods is very important for understand the real contribution.
There are acronym that are disclosure only after used in the manuscript.
The section Materials and Methods needs and important improvement in order to explain the adopted data, why these data are important and describe in details the adopted data. Regarding methods, need to be explained and justified the adopted options and its parameters.

Experimental design

Regarding the experimental design, the adopted data was not explained, its properties, how many features are available and why the adopted data are important.
Regarding feature selection methods, it was performed in order to find 50 better features for classification, why 50 features was adopted? What were the selected features?
The feature selection methods are dimensionality reduction approaches, however there are presented some results based on performance. What was the classifier used? What was the performance measure used? These information must be explained and justified in the manuscript.
By considering the Pattern Recognition research field, the one-class and two-class classification tasks are simpler problems. The authors could improve the discussion section in order to explain why the one-class and two-class is important in microRNAs classification task.

Validity of the findings

The is not possible to ensure that adopted data are robust, statistically sound, and controlled, considering that were not discussed/presented by the authors.
The conclusions are based in experimental results. However, the one-class and two-class classification is relatively simple in Pattern Recognition research field, in terms of methodology. In this way, the manuscript conclusions is very limited and inconclusive, regarding the adopted data, the chosen methods and the experiments design.
Besides, there is a mention by the authors: "Our SFC feature selection methodology". However, the manuscript do not present any new methodology, just a comparison among some feature selection methods.

---

## Round 0.2 · Minor Revisions

Please correct the typo and some sentences based on the reviewer's comment.

Reviewer 1 ·

Basic reporting

No comments.

Experimental design

No comments.

Validity of the findings

No comments.

Additional comments

The authors have made the corrections pointed in the first round of reviews.

In the new paragraph added to the Conclusion section, there are confusing sentences or typos that should be corrected:

- "For OCC feature selection nothing has been done in the area of pre-miRNA detection while one study investigated feature selection of OCC based mature miRNA prediction [37]. " I suggest rewriting it as "... while one study investigated feature selection based on OCC for mature miRNA prediction [37]."

- "...and compared to principal compoment analysis (PCA),..." should be 'component' instead of 'component'

-"These competing methods using differnet strategies", should be 'different' instead of differnet'

-"...in pre-miRNA detection don’t refer to OCC.". I suggest rewriting it as "in pre-miRNA detection and do not refer to OCC."

- "both of which was not done in previous appraoches." Rewrite as "both of which were not dot in previous approaches.".

---

## Round 0.3 · accepted · Accept

The authors revised correctly based on the reviewers' comments.